# Hypersexual Behavior and Depression Symptoms among Dating App Users

**Giacomo Ciocca** [1,*], **Lilybeth Fontanesi** [2], **Antonella Robilotta** [3], **Erika Limoncin** [4], **Filippo Maria Nimbi** [1], **Daniele Mollaioli** [4], **Andrea Sansone** [4], **Elena Colonnello** [4], **Chiara Simonelli** [1], **Giorgio Di Lorenzo** [5] and **Emmanuele A. Jannini** [4]

[1] Department of Dynamic and Clinical Psychology, and Health Studies, Sapienza University of Rome, 00185 Rome, Italy; filippo.nimbi@uniroma1.it (F.M.N.); chiara.simonelli@uniroma1.it (C.S.)

[2] Department of Psychological, Health and Territorial Sciences, University G. D'Annunzio of Chieti-Pescara, 66100 Chieti, Italy; lilybeth.fontanesi@unich.it

[3] Department of Human Sciences, University of L'Aquila, 67100 L'Aquila, Italy; antonellarobilotta96@gmail.com

[4] Chair of Endocrinology and Medical Sexology (ENDOSEX), Department of Systems Medicine, University of Rome Tor Vergata, 00133 Rome, Italy; erika.limoncin@gmail.com (E.L.); daniele.mollaioli@gmail.com (D.M.); andreasansone85@gmail.com (A.S.); elena_colonnello@hotmail.it (E.C.); eajannini@gmail.com (E.A.J.)

[5] Chair of Psychiatry, Department of Systems Medicine, University of Rome Tor Vergata, 00133 Rome, Italy; di.lorenzo@med.uniroma2.it

\* Correspondence: giacomo.ciocca@uniroma1.it

**Abstract:** The use of Dating Applications (DAs) is widespread, and in some cases could be associated with psychosexological issues. Hence, we decided to investigate hypersexual behavior and depression symptoms among DA users and non-users. We recruited a snowball convenience sample of 1000 subjects through an online platform in 2020 for a cross-sectional study. One hundred and nineteen (11.9%) were classified as DA users and 881 as non-users. All subjects completed a sociodemographic questionnaire, the Hypersexual Behavior Inventory (HBI), to assess hypersexuality, and the Patient Health Questionnaire (PHQ-9) to evaluate depression symptoms. We primarily found higher levels of hypersexual behavior and depression symptoms in DA users compared to non-users. In particular, HBI Total (users = 42.27 ± 16.37 vs. non-users = 31.85 ± 12.06; $p < 0.0001$), HBI Coping (users = 17.92 ± 8.01 vs. non-users = 13.52 ± 6.03; $p < 0.0001$), HBI Control (users = 14.73 ± 6.68 vs. non-users = 10.71 ± 4.95; $p < 0.0001$), HBI Consequences (users = 9.62 ± 4.33 vs. non-users = 7.60 ± 3.37; $p < 0.0001$), PHQ-9 Depression (users = 12.82 ± 6.64 vs. nonusers = 10.05 ± 5.84; $p < 0.0001$). On the whole, we found that hypersexual behavior and depression symptoms strongly characterize DAs users; this evidence could represent an important associated factor in DA use for sexual purposes.

**Keywords:** hypersexual behavior; depression symptoms; dating app; sexual behavior; psychopathology

## 1. Introduction

Dating Applications (largely known as "dating apps" or DAs) constitute an example of modern technology developed with the aim to increase the possibility of finding a new partner through the internet. DAs are cheap, user-friendly, and associate the daily use of computers and, mainly, smartphones with the desire to find a partner, mostly but not exclusively for sexual purposes, that is for sexual activity in a short-term relationship [1]. The wide diffusion of DAs recently called many researchers to investigate this phenomenon from a psychosexological point of view [2,3]. For instance, some evidence was found about the psychosexological profile of DA users, mainly characterized by the so-called dark personality traits, i.e., Machiavellianism, narcissism, and psychopathy [4,5]. However, some studies also identified positive aspects of DA users; for example, a recent study interestingly showed the positive and healthy aspects of the use of Tinder, investigating the Light

Triad of personality, i.e., faith in humanity, humanism, and Kantianism [6]. On the other hand, if peculiar personality aspects characterize some DA users, a recent investigation also highlighted the high prevalence of major depression, anxiety, and general distress in young people using DAs [7]. In some cases, these psychological issues negatively involve the perception of self through lower levels of self-esteem, and impairment of body image and satisfaction [8]. There are other pieces of evidence in regard to *sensu stricto* sexual behavior and related aspects, such as: safe sex, sexual health and sexually transmitted diseases (STDs), marital infidelity, casual sex, and DAs used for long- or short-term relationships [9,10]. In all these cases, differences were highlighted based on gender and sexual orientation, identifying peculiar aspects of sexual behavior, mating strategies, and risky sexual behavior among DA users [2]. For instance, a few controversial pieces of evidence regard the use of condoms and contraceptive methods related to DAs [2,11]. Hence, overall, the literature has investigated well the sexual behavior, personality traits, as well as mental health, of DA users, but there is a lack of knowledge on the reciprocal relationship between problematic sexuality and mental health. In fact, hypersexuality is characterized by excessive sexual activity, sexual obsessiveness, compulsive drives, and other behavioral aspects recalling addiction disorders, dysregulation of impulses involving the reward system related to the pleasure experience, and also obsessive-compulsive phenomenology in an attempt to cope with internal suffering [12–14]. On the other hand, hypersexual subjects try to alleviate depressive states and dysphoric moods through sexual activity, although in a dysregulated way [14,15]. Moreover, the relationship between hypersexuality and depression has been well documented in different study samples in an interesting metanalytic article [16,17]. In this regard, depression is conventionally characterized by a depressed mood, loss of interest or pleasure, sadness, irritation, and other symptoms that specify different clinical and subclinical forms of depressive disorders as major depression or dysthymia [18,19]. Depression is related in several ways to hypersexuality [15,20–23], and these two conditions could push some vulnerable subjects to use DAs, as also evidenced by recent articles discussing the relationship between internet use for dating purposes, sexual compulsivity, and psychological problems like depression, anxiety, and obsessive-compulsive symptoms [24,25]. Therefore, we have hypothesized two psychopathological phenomena, mutually correlated, that could influence and characterize the use of DAs: hypersexuality and depression. We have chosen these two particular facets of the psychopathology of sexual behavior for specific reasons regarding the addictive attitude that could characterize the dysfunctional use of DAs [24]. Moreover, these psychological and behavioral traits could secondarily condition safe sex and the use of contraceptive methods among DA users [11]. Based on the above-mentioned considerations, and according to our previously mentioned hypotheses, we firstly aimed to analyze the levels of hypersexual behavior and depression symptoms in people using DAs.

## 2. Materials and Methods

### 2.1. Recruitment

Through an online platform (Google Form) shared by the main social networks (Facebook, LinkedIn, WhatsApp), a convenience sample consisting of 1000 subjects was recruited through snowball recruitment (females and males: 18–60 years of age). According to our aims, the study design was observational and cross-sectional. This recruitment and collection of data was part of a more extensive study conducted in 2020 on hypersexual behavior among a large Italian sample. The recruited individuals completed a sociodemographic form, and were subjected to psychometric tests to assess hypersexual behavior and depression symptoms. Then, the total sample was divided into DA users and non-users in order to compare our outcome measures (specific item: <<Do you use DAs?>>).

Inclusion criteria were a self-disclosure not to be in treatment for severe mental issues, and participants must have reached the age of maturity. All participants were informed about the research and did not receive any financial remuneration for their participation in this study. Each participant, therefore, gave their own consent for the study by responding

to a specific item on the online platform. The entire protocol was anonymous, and the ethics committee of the Department of Dynamic and Clinical Psychology, and Health Studies approved this protocol (protocol number: 0000188, date 17 February 2020).

*2.2. Measures*

2.2.1. Sociodemographic Characteristics and DA Use

Basic sociodemographic information about gender, age, education, nationality, relational status, sexual orientation, contraceptive methods, type of sexual relationship (stable/casual), and gender of partners (male, female, both) was collected. Moreover, all participants indicated whether or not they used a DA account (YES/NO), and also which DA (Tinder, Bumble, Grindr, etc.). This above-mentioned information was collected by means of a specific sociodemographic questionnaire.

2.2.2. Hypersexual Behavior

Hypersexual behavior was evaluated using the Hypersexual Behavior Inventory (HBI). The HBI comprises 19 items along a 5-point Likert scale, from 1 = "Never" to 5 = "Very often". HBI assesses hypersexuality via three factors. The "coping" factor (seven items) assesses sex and sexual behaviors as a response to emotional distress, such as sadness, restlessness, or daily life worries. The "control" factor (eight items) assesses the lack of self-control in sexuality-related behaviors, such as an individual's attempt to change his or her sexual behavior failures. The "consequences" factor (four items) assesses the various consequences of sexual thoughts, urges, and behaviors, such as sexual activities that interfere with educational or occupational duties [26,27]. According to the HBI cut-off score identified by Reid et al. [27], we considered scores equal or higher to 53 as problematic and at risk for hypersexual behavior. The Italian version of HBI was used. HBI is characterized by a Cronbach's alpha coefficient of 0.95 [28].

2.2.3. Depression Symptoms

Depression symptoms were evaluated by administering the Patient Health Questionnaire (PHQ-9). The PHQ-9 consists of nine items along a four-point Likert scale, from 0 = "not at all" to 3 = "nearly every day". It is currently one of the most validated psychometric instruments in assessing mental health, and can be a powerful tool in assisting clinicians with diagnosing depression and monitoring its treatment [29,30]. It shows different levels of cut-off scores as follows: 0–4 = minimal or none; 5–9 = mild; 10–14 = moderate; 15–19 = moderately severe; 20–27 = severe. In this study, we used the scores $\geq$20 to identify individuals with severe depression symptoms, and the interval scores 15–19 and 10–14 for moderately severe and moderate depression symptoms, respectively, with a Cronbach's alpha coefficient of 0.89. The Italian version of the PHQ was used [30].

2.2.4. Sexual Orientation

To assess sexual orientation, the Kinsey Scale was used [31]. It is a self-reporting tool that assesses sexual behaviors and interests. Scores for this scale range from 0 = "exclusively heterosexual" to 6 = "exclusively homosexual". Scores from 1 to 5 identify individuals with different levels of same- and other-sex attraction and sexual behavior. An additional version of the Kinsey Scale includes an additional "X" category for those who do not fit within the 0 to 6 continuum. The "X" category is intended to describe "asexuality" or individuals who identify as "non-sexual" [31,32]. Moreover, to perform our analyses, we developed a further bi-categorization based on point zero of the Kinsey Scale (exclusively heterosexual) identifying heterosexuality, and points from 1 to 6 of the Kinsey Scale identifying non-heterosexuality [33].

2.2.5. Statistical Analysis

Continuous variables were statistically represented as means and standard deviations (SDs). Variables of the two study groups were compared using the Two-Sample *t*-Test for

Equal Means with two-tailed significance. Moreover, Cohen's *d* coefficient was calculated to estimate the effect size in the pairwise comparisons. Dichotomic variables were represented statistically as absolute and percentage frequencies. The difference between categorical variables was tested using the Relative Risk (RR). A binary logistic regression model was used to investigate the role of gender (dummy variables: females = 0; males = 1), relational status (in a relationship = 0; single = 1), sexual orientation (heterosexuals = 0; non-heterosexuals = 1), severe depression symptoms (PHQ-9 < 20 = 0; PHQ $\geq$ 20 = 1), moderately severe depression symptoms (other values = 0; $15 \leq$ PHQ-9 $\leq 19 = 1$), moderate depression symptoms (other values = 0; $10 \leq$ PHQ-9 $\leq 14 = 1$) (hypersexual behavior (HBI < 53 = 0; HBI $\geq$ 53 = 1), and age (years > 40 = 0; years < 40 = 1) in DA use. Each alpha error lower than 5% indicated statistical significance. All tests included the two-tail test and were conducted using the software program IBM SPSS Statistics for Windows, version 24 (IBM Corp., Armonk, NY, USA).

## 3. Results

We found that almost 12% of the recruited population uses DAs, and that users are significantly younger than non-users (age: non-users = 29.97 $\pm$ 11.36 vs. users = 26.81 $\pm$ 6.37 years of age, *p* = 0.003) (Table 1). When we analyzed sexual behavior, we found a higher tendency to establish sexual relationships with casual partners among DA users than non-users (casual partner: non-users = 11.6% vs. users = 52.1%, *p* < 0.0001), regardless of the gender. Conversely, non-users are characterized by the presence of a stable partner (stable partner: non-users = 74.6% vs. users = 31.1%, *p* < 0.0001). No significant differences were found for contraceptive methods, except for more frequent condom use among DA users (Table 2). Moreover, as shown in Table 2, more different declinations of sexual orientation were found among DA users compared to non-users, according to the seven points of the Kinsey scale. On the whole, non-heterosexual behavior was observed in 47.9% of DA users, compared to 18.5% of non-users (Table 2).

**Table 1.** Demographics (sample *n* = 1000).

| | | Overall Sample *n* = 1000 | Non-Users *n* = 881; % = 88.1 | Users *n* = 119; % = 11.9 |
|---|---|---|---|---|
| **Age** | | Mean $\pm$ SD | Mean $\pm$ SD | Mean $\pm$ SD |
| | | 29.59 $\pm$ 10.94 | 29.97 $\pm$ 11.36 | 26.81 $\pm$ 6.37 * |
| | | *n*; % | *n*; % | *n*; % |
| **Gender** | Females | 711; 71.1 | 639; 72.6 | 72; 60.5 |
| | Males | 289; 28.9 | 242; 27.4 | 47; 39.5 |
| **Relational Status** | In a relationship | 663; 66.3 | 626; 71.1 | 37; 31.1 |
| | Single | 337; 33.7 | 255; 28.9 | 82; 68.9 |
| **Education Level** | Secondary | 29; 2.7 | 27; 3.1 | 2; 0.8 |
| | High Secondary School | 448; 44.8 | 403; 45.7 | 45; 37.8 |
| | Bachelor's degree | 273; 27.3 | 232; 26.3 | 41; 34.5 |
| | Master's Degree | 169; 16.9 | 144; 16.3 | 25; 21.0 |
| | Post-graduate Degree | 81; 8.1 | 74; 8.4 | 7; 5.9 |

* A statistically significant difference between non-users and users is found (Student *t*-test: *p* = 0.003); SD = standard deviation.

Hence, Table 3 describes the psychological issues surrounding hypersexual behavior and depression symptoms among users and non-users of DAs. Several statistically significant differences emerged between these two groups. DA users showed higher levels of hypersexual behavior (HBI Total: non-users = 31.85 $\pm$ 12.06 vs. users = 42.27 $\pm$ 16.37; *p* < 0.0001; Cohen's d = 0.724). Moreover, in the three domains of HBI, DA users also showed statistically significant higher scores compared to non-users (Table 3).

**Table 2.** Sexual behavior in DA users and non-users.

| | | Non-Users 881 | Users 119 | RR; *p*-Value |
|---|---|---|---|---|
| | | *n*; % | *n*; % | |
| **Sexual Relationship** | No partner | 122; 13.8 | 20; 16.8 | 1.21; 0.38 |
| | Casual partner | 102; 11.6 | 62; 52.1 | 4.5; <0.0001 |
| | Stable partner | 657; 74.6 | 37; 31.1 | 0.47; <0.0001 |
| **Gender of Sexual Partner** | Both Genders | 53; 6.0 | 17; 14.3 | 2.37; 0.0009 |
| | Female | 225; 25.5 | 33; 27.7 | 1.08; 0.6 |
| | Male | 603; 68.4 | 69; 58.0 | 0.84; 0.04 |
| **Contraceptive Methods** | Coitus Interruptus | 87; 9.9 | 5; 4.2 | 0.42; 0.06 |
| | Implantable Uterine Device | 7; 0.8 | 2; 1.7 | 2.1; 0.34 |
| | Natural | 6; 0.7 | 0; 0 | n.a. |
| | None | 235; 26.7 | 24; 20.2 | 0.75; 0.14 |
| | Hormonal | 146; 16.6 | 13; 10.9 | 0.65; 0.12 |
| | Condom | 398; 45.2 | 75; 63.0 | 1.39; <0.0001 |
| | Sterilization | 2; 0.2 | 5; 4.2 | 18.5; 0.0004 |
| **Kinsey Scale** | Exclusively heterosexual | 718; 81.5 | 62; 52.1 | 0.63; <0.0001 |
| | Predominantly heterosexual, only incidentally homosexual | 58; 6.6 | 6; 5.0 | 0.76; 0.63 |
| | Predominantly heterosexual, but more than incidentally homosexual | 17; 1.9 | 8; 6.7 | 3.48; 0.002 |
| | Equally heterosexual and homosexual | 43; 4.9 | 19; 16.0 | 3.27; <0.0001 |
| | Predominantly homosexual, but more than incidentally heterosexual | 8; 0.9 | 2; 1.7 | 1.85; 0.43 |
| | Predominantly homosexual, only incidentally heterosexual | 6; 0.7 | 10; 8.4 | 12.33; <0.0001 |
| | Exclusively homosexual | 26; 3.0 | 12; 10.1 | 3.41; 0.0002 |
| | Asexual | 5; 0.6 | 0; 0 | n.a. |
| **Sexual orientation** | Heterosexual | 718; 81.5 | 62; 52.1 | 0.63; <0.0001 |
| | Non-heterosexual | 163; 18.5 | 57; 47.9 | 2.58; <.0001 |

RR: Relative Risk; n.a.: not applicable.

We found a statistically significant higher prevalence of subjects with HBI $\geq$ 53 among users compared to non-users (non-users = 7% vs. users = 27.7%; $p < 0.0001$; RR = 3.94).

In addition, depression symptoms mostly characterized DA users compared to non-users, according to the different cut-off scores: (PHQ-9 $\geq$ 20: non-users = 7.8% vs. users = 16%; $p = 0.0003$; RR = 2.03); ($15 \leq$ PHQ-9 $\leq$ 19: non-users = 13.6% vs. users = 27.7%; $p = 0.0001$; RR = 2.03).

Moreover, PHQ-9 levels (non-users = 10.05 $\pm$ 5.84 vs. users = 12.82 $\pm$ 6.64; $p < 0.0001$; Cohen's d = 0.443) (Table 3). Lastly, binary logistic regression explained the 25% variance (R2Nagelkerke = 0.25), and revealed the following sociodemographic, sexological and psychological characteristics associated with DA use: hypersexual behavior (Exp(B) = 2.414; $p = 0.002$), severe depression symptoms (Exp(B) = 2.227; $p = 0.02$), moderately severe depression symptoms (Exp(B) = 2.162; $p = 0.009$), male gender (Exp(B) = 1.760; $p = 0.015$), under 40 years of age (Exp(B) = 2.236; $p = 0.047$), single relationship status (Exp(B) = 4.577; $p < 0.0001$), and non-heterosexuality (Exp(B) = 3.298; $p < 0.0001$) (Table 4).

**Table 3.** Hypersexual Behavior and Depression Symptoms in DA users and non-users.

| | | Non-Users | Users | *p*-Value | Effect Size Cohen's d |
|---|---|---|---|---|---|
| **Hypersexual Behavior** | HBI Total mean ± sd | 31.85 ± 12.06 | 42.27 ± 16.37 | 0.0001 * | 0.724 |
| | HBI Coping | 13.52 ± 6.03 | 17.92 ± 8.01 | 0.0001 | 0.620 |
| | HBI Control | 10.71 ± 4.95 | 14.73 ± 6.68 | 0.0001 | 0.683 |
| | HBI Consequences | 7.60 ± 3.37 | 9.62 ± 4.33 | 0.0001 | 0.520 |
| | HBI ≥ 53 [1] *n*; % | 62; 7 | 33; 27.7 | 0.0001 ** | / |
| **Depression Symptoms** | PHQ-9 mean ± sd | 10.05 ± 5.84 | 12.82 ± 6.64 | 0.0001 | 0.443 |
| | PHQ-9 ≥ 20 [1] *n*; % | 69; 7.8 | 19; 16 | 0.0003 *** | / |
| | 15 ≤ PHQ-9 ≤ 19 [2] *n*; % | 120; 13.6 | 33; 27.7 | 0.0001 **** | / |
| | 10 ≤ PHQ-9 ≤ 14 [3] *n*; % | 229; 26 | 27; 22.7 | ns | / |

* Two-Sample *t*-Test for Equal Means with two-tailed significance; ** RR = 3.94; *** RR = 2.03; **** RR = 2.03; [1] severe; [2] moderately severe; [3] moderate; ns = not significant; HBI: Hypersexual Behavior Inventory; PHQ: Patient Health Questionnaire; RR: Relative Risk.

**Table 4.** Binary logistic regression about hypersexual behavior, depression symptoms, and sociodemographic aspects in DA users.

| | B | S.E. | Wald | gl | Sign. | Exp(B) |
|---|---|---|---|---|---|---|
| **Hypersexual Behavior** (HBI ≥ 53 = 1) * | 0.881 | 0.291 | 9.156 | 1 | 0.002 | 2.414 |
| **Severe Depression Symptoms** (PHQ-9 ≥ 20 = 1) * | 0.800 | 0.360 | 4.947 | 1 | 0.026 | 2.227 |
| **Moderately Severe Depression Symptoms** (15 ≤ PHQ-9 ≤ 19 = 1) * | 0.771 | 0.294 | 6.864 | 1 | 0.009 | 2.162 |
| **Moderate Depression Symptoms** (10 ≤ PHQ-9 ≤ 14 = 1) * | 0.120 | 0.286 | 0.176 | 1 | 0.675 | 1.127 |
| **Gender** (male = 1) | 0.566 | 0.232 | 5.947 | 1 | 0.015 | 1.760 |
| **Age** (Age < 40 = 1) | 0.805 | 0.404 | 3.957 | 1 | 0.047 | 2.236 |
| **Relational status** (single = 1) | 1.521 | 0.225 | 45.881 | 1 | 0.000 | 4.577 |
| **Sexual Orientation** (non-heterosexuals = 1) | 1.193 | 0.226 | 27.829 | 1 | 0.000 | 3.298 |

* cut-off scores; $R^2_{Nagelkerke}$ = 0.26; HBI: Hypersexual Behavior Inventory; PHQ: Patient Health Questionnaire

## 4. Discussion

In this study, we found significant associations between particular aspects of psychological suffering and the use of DAs. In particular, depression symptoms, together with hypersexual behavior, mostly characterize some DA users in comparison with non-users. It is also possible to consider the role of DAs as a "transitional object", providing security, emotional well-being, and symbolic connection with a valued other, for some users attempting to cope with their internal depressive feelings by having sex in a dysfunctional way [34,35]. On the other hand, the association between hypersexual behavior and depression was documented, also concerning online sexual activities, such as cybersex or the use of pornography [16,17]. Therefore, it is plausible to hypothesize the primary role of a dysregulated attitude towards sexual behavior of some DA users, which is manifested through depressive and hypersexual symptoms, as demonstrated here. The statistically robust association between depression symptoms and hypersexual behavior in the use of DAs represents a novelty in this field. The higher scores in depression symptoms and hypersexual behavior, together with a larger percentage of prevalence of these two phenomena among DA users, are the most interesting findings we were able to demonstrate. In users of DAs, we found higher levels of depression symptoms compared to non-users, and this evidence ought to be under consideration during the screening phase or research protocol on DAs, especially among young adults. In particular, feeling down, the loss of

pleasure, sleeping and eating disorders are only some of the symptoms characterizing our evaluation of depression states in users of DAs [29]. The relationship between life instinct and death anxiety was discussed in light of pathological issues such as depression and hypersexuality [15,21,36]. This controversial link, first introduced by Freud in his work entitled *Beyond the pleasure principle*, seems particularly true in some of the more fragile users of DAs when they fight internal depressive suffering with a compulsive search for sex and casual partners [23,37]. In this regard, we recall that the use of DAs is often related to sexual purposes and casual sex, as the descriptive data of our study have also revealed, where people with a casual partner are significantly more numerous among users of Das [38,39]. Moreover, the higher scores of hypersexuality and its domains as assessed by HBI, i.e., coping, control, and consequences, together with the worst score of depression symptoms in users of Das compared to non-users, revealed, for the first time, a close link between depression and an excessive search for sexual activity among DA users. On the whole, we found a new connection between hypersexual behavior, depression symptoms, and use of DAs, such that it is possible to speculate about the direction of this connection. Our regression model, explaining about 26% of the variance, revealed the importance of sociodemographic characteristics, such as non-heterosexual orientation, being single, male and under 40 years of age, but hypersexuality and depression also play an important role in this explicative model related to DA use. Based on this evidence concerning sociodemographic data, it is plausible to hypothesize a greater attitude of hypersexual behaviors, together with a higher presence of depressive feelings, in single and non-heterosexual people using DAs [40,41]. More generally speaking, DAs could indirectly represent caring tools to alleviate depressive states and problematic sexuality in more vulnerable individuals with a likely emotional dysregulation [42,43]. From this point of view, it is also increasingly necessary to implement and consider DAs for campaigns aimed at the prevention of psychological and sexological problems, as suggested by part of the literature [10]. On the other hand, some investigations demonstrated that insecure and problematic personalities characterized by low self-esteem and an insecure attachment style are associated with DA use [44,45]. In particular, lack of self-esteem and the subsequent search of self-esteem enhancement could be hypothetically related to both hypersexuality and depression in the problematic use of DAs [44]. Hence, besides the strong association between depression symptoms and hypersexual behavior among DA users, we found that specific sociodemographic characteristics of our recruited population are more related to the use of DAs. These aspects partially support social differences previously demonstrated in DA use according to gender, sexual orientation, and relationship status [5,46–48]. Moreover, they represent a new useful point of view to better define the psychological and social profile of typical users of DAs. Furthermore, no differences in terms of use of contraception and safe sex between DA users and non-users were found, as previously demonstrated in a comparable sample mainly composed of heterosexual women [11]. Conversely, we interestingly found a higher presence of condom use among DA users, likely due to a greater tendency towards casual partners and casual sex compared to non-users. Therefore, according to the information provided here, it is also possible to modulate the false stereotype associating the use of DAs to unprotected sex. This aspect is important in terms of public health and demonstrates that DA users are more predisposed to safe sex, and are also more likely to be indicated as a target for peer sex education concerning the use of condoms [2]. This study suffers from a few limitations due to the unbalanced gender distribution of the sample, though similar to most recruitment in convenience samples [33,49]. In addition, some lacking aspects related to relationship status, such as polyamory and open relationships, could represent a limit, together with the cross-sectional design of the study and the snowball recruitment. As a future aim, we plan on reducing the current study limitations by investigating the subgroups of our sample in a more specific manner according to sociodemographic characteristics, in particular single people. Moreover, a clinical study on patients specifically diagnosed and treated for depression or hypersexuality could reinforce our findings, mainly based on the limits of our observational study. Finally, a standardized methodology to assess DA use is suitable,

following the frequency and the motivation to DA use, but few tools and questionnaires are available, especially those with linguistic validation. This is another limit of this study that it is necessary to improve in a future study.

## 5. Implications

The risk of hypersexual and depression symptoms in DA users is higher compared the general population. This evidence is relevant for clinicians assessing and treating young people. In these cases, when a clinical psychologist, a psychiatrist or an educator assesses a young patient for mental problems, he should also take into consideration some behavioral habits, such as the use of smartphone applications for dating.

## 6. Conclusions

This investigation shows for the first time a strong association between DA use, hypersexual behavior, and depression symptoms. This evidence could mean that some hypersexual and/or depressed individuals recurringly use DAs to alleviate their psychological and sexological suffering. Therefore, we should also consider DAs as technological tools used by some individuals in order find a possible solution to their psychosexological issues, for sexual and relational reasons. Finding a partner or a sexual partner, including through a DA, might always be considered a positive and healthy behavior on the part of a subject attempting to cope with an internal state of frustration, dissatisfaction, or psychosexological suffering. In light of these considerations, it would be appropriate to consider DAs as a place to eventually promote psychological, relationship and sexological health.

**Author Contributions:** Conceptualization, G.C. and E.A.J.; methodology, G.C., A.S. and D.M.; investigation, A.R., G.C. and L.F.; data curation, G.C., L.F. and A.R.; writing—original draft preparation, G.C.; writing—review and editing, E.L., F.M.N. and E.C.; supervision, E.A.J., G.D.L. and C.S. All authors have read and agreed to the published version of the manuscript.

**Funding:** This research received no external funding.

**Institutional Review Board Statement:** The study was conducted in accordance with the Declaration of Helsinki, and approved by the Ethics Committee of Department of Dynamic and Clinical Psychology and Health Studies of Sapienza University of Rome (protocol code n. 0000188 17/02/2020)." for studies involving humans.

**Informed Consent Statement:** Informed consent was obtained from all subjects involved in the study.

**Data Availability Statement:** The data presented in this study are available on request from the corresponding author.

**Conflicts of Interest:** The authors declare no conflict of interest.

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
