# Peer review of "Hypersexual Behavior and Depression Symptoms among Dating App Users"

_sexes, doi:10.3390/sexes3020023_

Round 1

Reviewer 1 Report

None of the questions raised in the previous review have been adequately addressed.

Author Response

Reviewer 1

None of the questions raised in the previous review have been adequately addressed.

Authors

We thank the reviewer.

Reviewer 2 Report

  1. More references on DA, hypersexuality and depression should benefit the soundness of the introduction.
  2. Please write clearly all the objectives of the investigations (lines 83-86).
  3. Wasn’t intensity of DA use measured? How often, for how many hours, etc?
  4. If sexual orientation is presented as a major measure, how come it wasn’t operationalized in the introduction nor in the objectives?
  5. Discussion: Please provide paragraphs. Please include an implications sub section.

Best wishes.

Author Response

Reviewer 2

More references on DA, hypersexuality and depression should benefit the soundness of the introduction.

Authors

We thank the reviewer for this comment, but the references about this topic are very few. We have inserted the references and study actually present in literature. In this regard, we consider our article original.

Reviewer 2

Please write clearly all the objectives of the investigations (lines 83-86).

Authors

We thank the reviewer for this suggestion and we wrote more clearly the objectives as follow: “we firstly aimed to investigate the levels of hypersexual behavior and depression symptoms in people using DAs”.

Reviewer 2

Wasn’t intensity of DA use measured? How often, for how many hours, etc?

Authors

We are in accord with the reviewer about these questions, but we just affirmed this aspect as a study limitation in the discussion as follow: “Finally, a standardized methodology to assess the DA use is suitable, following the frequency and the motivation to DA use, but few tools and questionnaire are available also a linguistic validation. This last is another limit of this study that it is necessary to improve in a future study”.

Reviewer 2

If sexual orientation is presented as a major measure, how come it wasn’t operationalized in the introduction nor in the objectives?

Authors

We thank the reviewer, but we considered sexual orientation into the sociodemographic information as gender, age, and so on. Therefore, in the objective it is present when we speak about “also controlling some sociodemographic aspects”

Reviewer 2

Discussion: Please provide paragraphs. Please include an implications sub section.

Authors

We thank the reviewer for this useful suggestion, we have added a specific sub-section with the folowing paragraph of implication: The risk of hypersexual and depression symptoms in DA users is higher compared the general population. This evidence is relevant for clinicians assessing and treating young people. In these cases, when a clinical psychologist, a psychiatrist or an educator assess a young patient for mental problem, he should take into consideration also some behavioral habit, as the use of smartphone application for dating.”

Reviewer 2

Best wishes.

Authors

We thank the reviewer.

Reviewer 3 Report

I would like to thank for the opportunity to review your work and congratulate you for choosing the interesting topic of the article. The manuscript is important to the practice of Psychology and Sexology and shows interesting association between dating apps, sexual health and depressive symptoms.

The methodology is appropriate for the objectives of the study but there is no information when the study has been conducted? Inclusion/Exclusion criteria should be stated. Also, respondents should be asked about previous depressive episodes and therapy. The results are clearly presented. The discussion explains the implications of the results. Authors accurately discussed the limitations. I suggest to recruit bigger sample of DA users for further studies for better groups comparison.

Author Response

Reviewer 3

The methodology is appropriate for the objectives of the study but there is no information when the study has been conducted? Inclusion/Exclusion criteria should be stated. Also, respondents should be asked about previous depressive episodes and therapy.

Authors

We than the reviewer, we have added the follow sentence about this suggestion: “Inclusion criteria were a self-disclosure to be not in treatment for severe mental issues and must have reached the age of majority”. About the history of depression illnesses, we did not ask, but we have mentioned this aspect in the limitation section.

Reviewer 3

The results are clearly presented. The discussion explains the implications of the results. Authors accurately discussed the limitations. I suggest to recruit bigger sample of DA users for further studies for better groups comparison.

Authors

We thank the reviewer for this consideration. We have specified this in the limitations section.

Reviewer 4 Report

I appreciate the opportunity to review this interesting study. 
I have not been able to study the level of plagiarism in the article, so I expect that the publisher and editor should do so, and not the reviewer. 

It is possible that there are several citations from the same author who is the main author of the study, the degree of self-citation of the manuscript should be assessed. It is not a high number, 5 I thought I found. Incidentally, for a descriptive observational study the number of references is high. I have not assessed other authorship of the study in the citations. 

Considerations, 
1. you should include the study design and year of the study in the abstract. 

2. 

The introduction is interesting, but in my opinion not optimally structured, with only 2 paragraphs. I would advise to clearly divide the introduction into paragraphs as follows: 
definition of the problem dimension of the problem (there are no epidemiological data, none, to give us an idea of the size of the problem in society, despite the fact that 24 references are provided for an introduction that is not long), background, justification, and objective. 

3. As for the objective, I think it would be necessary to reformulate it to make it clearer what they intend, for example, they propose that the objective of this research is to investigate... I do not think that the verb investigate is the best option, they should replace it with a more operative and representative verb (to analyse, to determine...).

4.methodology: I consider that the first section that should appear is the design, year and scope of the study. As in the scope, they do not make it clear that this is an observational, descriptive and cross-sectional study (the latter appears in the objective of the study, which is inappropriate and should be eliminated there, so that it appears in the methodological design). 

One of the main problems with this type of online survey is the control of responses. I have not been able to check whether any kind of filter was used to avoid multiple responses from the same user. 
If this was not done, it should be clearly stated in the limitations of the study, because regardless of the convenience sample, there was no exhaustive control of the responses, so reliability may suffer.

I think they should not say that "our ethics committee approved the study", but that the ethics committee of XXXXXXXXXXX approved the study with code (or registration) number XXXXXXXX (line 100).

5. Results: The results are very well described and the analysis is interesting. 
I think the most important thing is to organise the information a little better, placing each table after the quotation in the text. They would be better distributed and would facilitate better reading and comprehension. 

6. 

I think the writing style of the discussion is inappropriate. In fact, there is a single huge paragraph, with no full stop to separate and organise the information. There is a need to organise it better and make it easy for the reader. 
Unify the topics you discuss, and differentiate them in a timely manner. 
Moreover, at several points they repeat many results that have already been presented and could perhaps be simplified, giving more room for pure discussion.

I also consider that some important limitations have not been taken into account and should be made explicit in the limitations, such as the poor control over responses (in the case of not having used internet protocol filters, for example), and not recognising that the cross-sectional study design does not provide robust evidence or causality. 

Author Response

I appreciate the opportunity to review this interesting study.

I have not been able to study the level of plagiarism in the article, so I expect that the publisher and editor should do so, and not the reviewer.

Authors

We thank the reviewer for the first part of comment.

Reviewer 4

It is possible that there are several citations from the same author who is the main author of the study, the degree of self-citation of the manuscript should be assessed. It is not a high number, 5 I thought I found. Incidentally, for a descriptive observational study the number of references is high. I have not assessed other authorship of the study in the citations.

Authors

We did not understand the significance of this comment. We citied some our article of hypersexual behavior. The self-citation did not appear numerous, for our judge.

Considerations,

Reviewer 4

  1. you should include the study design and year of the study in the abstract.

Authors

We thank the reviewer, for this comment. This was made in the revised manuscript.

Reviewer 4

  1. The introduction is interesting, but in my opinion not optimally structured, with only 2 paragraphs. I would advise to clearly divide the introduction into paragraphs as follows:

definition of the problem dimension of the problem (there are no epidemiological data, none, to give us an idea of the size of the problem in society, despite the fact that 24 references are provided for an introduction that is not long), background, justification, and objective.

Authors

We thank the reviewer for this comment, but we prefer not modify the introduction also according to the style of journal. The background, justification and objective are presented in the text.

Reviewer 4

  1. As for the objective, I think it would be necessary to reformulate it to make it clearer what they intend, for example, they propose that the objective of this research is to investigate... I do not think that the verb investigate is the best option, they should replace it with a more operative and representative verb (to analyse, to determine...).

Authors

We thank the reviewer for this comment. We have rewritten the objective and we have changed the verb “investigate” with “analyze”.

Reviewer 4

4.methodology: I consider that the first section that should appear is the design, year and scope of the study. As in the scope, they do not make it clear that this is an observational, descriptive and cross-sectional study (the latter appears in the objective of the study, which is inappropriate and should be eliminated there, so that it appears in the methodological design).

Authors

We thank the reviewer and we have added the description of design in the method.

Reviewer 4

One of the main problems with this type of online survey is the control of responses. I have not been able to check whether any kind of filter was used to avoid multiple responses from the same user.

If this was not done, it should be clearly stated in the limitations of the study, because regardless of the convenience sample, there was no exhaustive control of the responses, so reliability may suffer.

Authors

We thank the reviewer for this comment. It is easy to check the duplicate of response. We have checked this aspect and some cases we have deleted the duplicates

Reviewer 4

I think they should not say that "our ethics committee approved the study", but that the ethics committee of XXXXXXXXXXX approved the study with code (or registration) number XXXXXXXX (line 100).

Authors

We thank for this comment we have added the protocol number 0000188 in the date 17/02/2020 and the name of ethics committee concerning Department of Dynamic and Clinical Psychology, and Health Studies.

Reviewer 4

  1. Results: The results are very well described and the analysis is interesting.

I think the most important thing is to organise the information a little better, placing each table after the quotation in the text. They would be better distributed and would facilitate better reading and comprehension.

Authors

We thank the reviewer for this comment. Also, in this case we have distributed the tables according to journal format.  

Reviewer 4

  1. I think the writing style of the discussion is inappropriate. In fact, there is a single huge paragraph, with no full stop to separate and organise the information. There is a need to organise it better and make it easy for the reader.

Unify the topics you discuss, and differentiate them in a timely manner.

Moreover, at several points they repeat many results that have already been presented and could perhaps be simplified, giving more room for pure discussion.

Authors

According to this comment we have inserted another paragraph about the implications of our study. The rest of discussion, instead, appears coherently organize among the major finding regarding the relationship between hypersexual behavior and depression and then between sexual behavior in DA users.

Reviewer 4

I also consider that some important limitations have not been taken into account and should be made explicit in the limitations, such as the poor control over responses (in the case of not having used internet protocol filters, for example), and not recognising that the cross-sectional study design does not provide robust evidence or causality.

Authors

We thank the reviewer for this comment. We have updated the limitation section according to this suggestion.

Round 2

Reviewer 2 Report

Thank you for implementing the requested changes. I believe the article is now fit for publication.

Best wishes.

This manuscript is a resubmission of an earlier submission. The following is a list of the peer review reports and author responses from that submission.

Round 1

Reviewer 1 Report

This is an interesting study that needs a stronger rationale in terms of its contribution to the study of hypersexuality and dating app use. The manuscript also needs to be theoretical framed and it is potentially bias by the way authors measured dating application use.  

Introduction:

I think that the paper did not set up a convincing delineation of the incremental contribution the study makes to the previous empirical literature. I am not saying there isn’t one there but just that it needs to be more clearly articulated in the context of an empirical study. Right now, the rationale for the study seems vague and guided by the fact that hypersexuality could be related with dating use app but we should bear in mind that maybe the different motives for using dating apps could be differently related with hypersexuality and depression. For example, hypersexuality could be more related with those dating users looking for hookups.

Authors should make a bigger effort to justify their study and explain what it offers to the existent literature. Author should also include the theoretical framework that guided their study. In fact, the short introduction reads a bit as if the authors just grabbed two variables potentially related with dating app use. I recommend that the authors reconsider framing their study in terms of theory to agglutinate all the variables selected in relation to nomophobia. It also seems that authors have split their research on several manuscript and we are missing the big picture. This could be problematic in terms of ethics in publishing.

The introduction should end with clearly articulated expectations (hypotheses) that advance current knowledge and that drive the data analysis. Hypothesis should also be framed in theoretical and empirical background. Authors should explained in what information is based each hypothesis.

Method:

Another problem is the insufficient description of the sample. Author/s should include a discussion of the desired sample based on a power analysis, then the procedure used (i.e., who was contacted about participation), and finally the number of participants who were involved in the study.

Was used an instructional manipulation check to verify that the participants had read the survey instructions and answer appropriately the questions in the online survey.

How long did the data collection process take overall?

What month was open the survey and what month was close the survey? How long was the survey posted online? What year was gathered the data?

The use of a single item to measure dating use it is problematic. First, we are missing frequency of use and second, we are missing motive to use dating apps. Different frequencies and motives can be differently related with hypersexual behavior and depression. From my point of view this is a serious flaw that can invalidate the results obtained. Authors should at least justify why they used a single item and how this flaw can affect results.

 Results:

I find no weaknesses in the statistical analyses conducted.

Discussion:

The discussion needs also to be frame in terms of theory (as Introduction).

A stronger call for future research is needed.

Reviewer 2 Report

The authors explore the associations between dating app use and a number of variables including depression, hypersexuality, sexual preferences, sexual activity, use of contraception, and stable partner relationship.  The topic is interesting and suitable for Sexes.

The main weakness of the paper is that the presentation of statistical analyses is inappropriate.  The authors should consider seeking help from a colleague who understands statistics and how to present the methods and results sections in a conventional and conceptually cohesive manner.  Below are listed examples of issues that need to be addressed.

Line 146  “Variances of the two study groups are compared using Two-Sample t-Test …”  How does the t-test compare variances?  Do the data meet the requirements for valid application of the t-test?

Lines 152-158: Here the authors list details of the dummy-variable coding strategy used with SPSS without explaining what these represent.  The authors should enlist the help of an expert who understands what these represent and rewrite this section.

Line 158 “Each alpha error lower than 5% …”  Is the term “alpha error” used correctly here?

Lines 162-163 “The primary finding of this study was that almost 12% of the recruited population uses DAs and that users are significantly younger than non-users.”  Was that the primary finding?  How do the authors reconcile this with the abstract, which reads “We primarily found higher levels of hypersexual behavior and depression symptoms in DA users compared to non-users.”  What is meant by a primary hypothesis and how does that relate to the need for a Bonferroni correction?  Have the authors considered applying such a correction?  Why or why not?  Should all of the analyses be considered exploratory and uncorrected for multiple comparisons?

Lines 174-175, Table 1: Where are the results for “the Chi-Square test or, when appropriate, the Fisher’s exact test” (as described on lines 150-151)?  Why is the t-test of differences in mean age the only test shown?

Lines 178-179, Table 2: Are the contraceptive methods categories mutually exclusive, or could a participant belong to more than one contraceptive category?

Tables 2 and 3: Does it make sense to compute the RR for all of the categorical variables?  For example what would be the relevance of comparing the RR for being moderately depressed with being less or more severely depressed?  Or more or less mixed in terms of sexual preference?  Would a chi-squared test be more appropriate?

Table 4: “Exp(B)” is how the output is labelled by SPSS, but is this term suitable for publication?  What does this term mean?